# Evaluating the Evaluators: On the Reliability of Automated Label Extraction for Radiology Reports

Raphael Stock[*]                                          RAPHAEL.STOCK@DKFZ-HEIDELBERG.DE
Moritz Langenberg[*]                              MORITZ.LANGENBERG@DKFZ-HEIDELBERG.DE
David Zimmerer[*]                                   DAVID.ZIMMERER@DKFZ-HEIDELBERG.DE
Katharina Dvornikovich              KATHARINA.DVORNIKOVICH@DKFZ-HEIDELBERG.DE
Julius C. Holzschuh                           JULIUS.HOLZSCHUH@DKFZ-HEIDELBERG.DE
Jonathan Suprijadi                          JONATHAN.SUPRIJADI@DKFZ-HEIDELBERG.DE
Constantin Ulrich                             CONSTANTIN.ULRICH@DKFZ-HEIDELBERG.DE
Aditya Rastogi                                               ADITYA.RASTOGI@UKBONN.DE
Kai Schlamp                                    KAI.SCHLAMP@MED.UNI-HEIDELBERG.DE
Philipp Vollmuth                                          PHILIPP.VOLLMUTH@UKBONN.DE
Klaus Maier-Hein                                   K.MAIER-HEIN@DKFZ-HEIDELBERG.DE

## Abstract

Automated extraction of structured clinical labels from radiology reports is a key evaluation method for vision-language models in medical imaging, yet the reliability of these labels is rarely questioned. We analyze agreement between multiple large language models (LLMs) and the labels from the CT-RATE dataset. Through our manual review of instances with inter-method disagreement, we identified errors in the CT-RATE labeling and subsequently manually corrected annotations. Furthermore, LLM-based annotators exhibit high labeling fidelity, while the RadBERT classifier, used to create the official labels for CT-RATE, shows higher error rates and degrades under distribution shift; specifically, the best LLM achieves a 60% reduction of CT-RATE's "ground truth" labeling errors. Beyond identifying the limitations of current labeling, we provide a solution for extracting reliable reference annotations, leading us to publish our refined CT-RATE labels at: https://zenodo.org/records/19597002.

**Keywords:** label extraction, radiology reports, large language models, evaluation

## 1. Introduction

Reliable extraction of structured labels from radiology reports is a fundamental requirement for developing and evaluating machine learning systems in medical imaging. These labels are widely used both for training predictive models and as reference annotations for benchmarking, yet their accuracy and robustness are rarely systematically assessed. For instance, in the large-scale CT-RATE dataset (Hamamci et al., 2024), the official labels were predominantly generated by a RadBERT-based classifier. However, the extractor's reliability, especially when applied to out-of-domain (OOD) data featuring varied reporting styles, was not reported. To address this gap, we investigate the reliability of large language models (LLMs) as universal training-free label extractors. We analyze the agreement between multiple LLMs using varying prompts and report augmentation and the RadBERT

---

*. Equal contribution. Co-first authors may list as lead on CV.

classifier. We demonstrate that while LLMs show strong overall agreement with RadBERT on the CT-RATE dataset, human expert annotation of contested instances reveals most LLMs consistently outperform RadBERT in-distribution. Furthermore, when applied to an external dataset with a different reporting style, the agreement between RadBERT and the LLMs degrades substantially, positioning LLMs as the potentially more reliable tool for varied reporting environments.

## 2. Methods

**Dataset.** We randomly sampled 200 radiology reports from the CT-RATE validation set, utilizing 18 classification labels to generate 3,600 label-case instances. We excluded *Lymphadenopathy* and *Lung opacity* due to their high semantic ambiguity. For the OOD evaluation, we used 882 chest CT reports from University Hospital Bonn.

**Automated LLM annotators.** Three prompt variants for GPT-OSS-120b (OpenAI, 2025) were used. A *Prompt Ensemble* is the majority vote of all three. A *Shuffle Ensemble* runs the same prompt with ten sentence permutations. Furthermore, *MiniMax-M1* (MiniMax AI, 2025), *MedGemma-1.5-4B* (Sellergren et al., 2026), and *Qwen3.5-120b* (Qwen Team, 2025) are open-weight LLMs used as additional independent annotators for comparison. Additionally, we utilize the CT-RATE fine-tuned *RadBERT* (Yan et al., 2022), the same classifier used to determine the official ground-truth labels for the CT-RATE dataset.

**Human adjudication.** Focusing on disagreements among four models (the three *Prompt Ensemble* models and *RadBERT*) identified 159 discrepancies across 101 cases, excluding the *Lymphadenopathy* and *Lung opacity* classes. Three human annotators then labeled these cases, achieving consensus on 97 instances; an expert radiologist reviewed the remaining 60 contested instances. Excluding six unclear instances resulted in a final cohort of 153 manually reviewed reference and 3041 automatic-model consensus instances.

## 3. Results

**Pairwise method agreement (Figure 1 a).** Against the reference labels (3 041 consensus instances + 153 expert-reviewed = 3 194 total), the models separate into two tiers: Qwen3.5 and the ensemble variants each disagree on fewer than 50 instances, whereas RadBERT, MiniMax, and MedGemma show progressively higher disagreement (82–238). This gap indicates that label-extraction quality varies substantially across methods and some selected LLMs may be more suited to label extraction than the widely deployed RadBERT model.

**Per-label failures (Figure 1 b).** The most error-prone classes, determined by median of per-model error rankings, are *Arterial wall calcification* (2), *Atelectasis* (3.5), and *Medical Material* (4). Qwen3.5 exhibits specific weaknesses in *Arterial wall calcification* (11 errors) and *Atelectasis* but remains the most reliable method overall. RadBERT's errors are more uniformly distributed across labels rather than concentrated on individual findings, suggesting that its failure modes differ qualitatively from those of the LLM-based annotators.

**Out-of-domain generalisation (Figure 1 c).** The agreement of GPT-OSS with RadBERT given by Cohens $\kappa$ drops from 0.92 (CT-RATE) to 0.59 (Bonn dataset), with per-label declines on most findings. Given that RadBERT was specifically trained on CT-RATE while GPT-OSS remains a general-purpose model, we hypothesize that RadBERT's perfor-

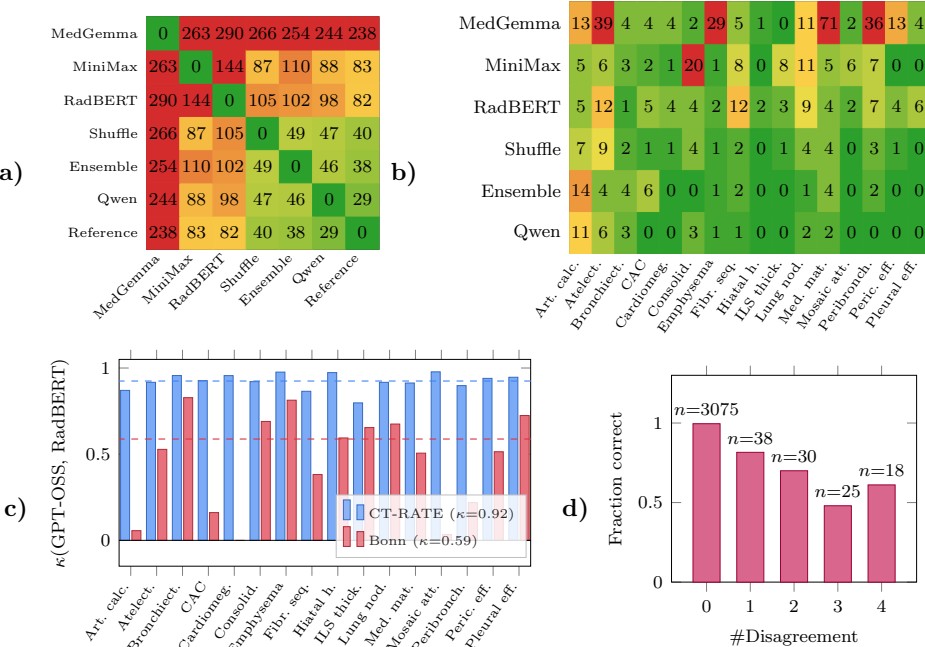

Figure 1: **a)** Pairwise disagreement counts (green=0, red=144; Reference: LLM & Rad-BERT consensus on 3041 easy pairs, human+expert on 153 hard pairs). **b)** Per-label disagreements vs. Reference labels for RadBERT, MiniMax, Ensemble, and Shuffle (sorted by total errors; Lymphadenopathy and Lung opacity excluded; green=0, red=20). **c)** Per-label Cohen's $\kappa$(GPT-OSS, RadBERT) on CT-RATE (blue) vs. Bonn (red), showing domain-specific degradation. **d)** Shuffle-ensemble calibration-fraction correct vs. reference labels by disagreement count.

mance degrades more significantly when applied to the OOD Bonn dataset. RadBERT's hypothesized lack of cross-dataset robustness likely accounts for the observed decrease in correlation. However, the exact source of degradation needs further examination.

**Ensemble uncertainty is reliable (Figure 1 d).** Total intra-*Shuffle Ensemble* agreement yields 99.6% accuracy against reference labels. However, when three or more models disagree, accuracy drops to 48–61%. Consequently, intra-ensemble disagreement serves as a valuable tool for separating trustworthy "easy instances" from complex ones.

## 4. Discussion and Conclusion

In-distribution, we observed 95% inter-method agreement. On the disputed 5%, Qwen3.5 and the ensemble methods align most accurately with expert assessments. Furthermore, general-purpose LLMs may prove more resilient to distribution shifts. Given sufficient computational resources, these results position current LLMs as a preferred solution for extracting reliable reference annotations essential for model training and benchmarking.

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
