# OpenReview forum: "Evaluating the Evaluators: On the Reliability of Automated Label Extraction for Radiology Reports"
_MIDL.io/2026/Short_Papers — MIDL 2026 - Short Papers Poster_

### Official Review · Reviewer_WN6k · 2026-04-29
**Questioning the reliability of automated label extracted from radiology reports**

**Rating:** 4
**Confidence:** 5

**Review:**

The paper is well written and easy to follow. Details of the annotation protocol are missing both in the Appendix and in the Zenodo repository. It focuses on data work, in particular on the importance on quality data annotations.

**Summary:**

The authors investigate the use of multiple large language models (LLMs) for label extraction on radiology reports. Through manual review and inter-method disagreement they identified errors in the CT-RATE labeling. They share their labels on Zenodo. They suggest that the best LLM might be a better choice than RadBert, which was used to generate most of the "ground truth" labels for the CT-RATE dataset.

**Strengths:**

- This paper identifies limitations of current labeling (RadBert).
- This work focuses on the quality of the data annotations instead of presenting a novel method.
- The authors analyze agreement both in-domain (CT-RATE) and out-domain (Bonn dataset).

**Weaknesses:**

- Limited details are provided about RadBERT.
- There is limited related work. The authors could consider reviewing similar studies on Chest X-ray datasets [1,2].
- The authors could consider further analysis, for example analyzing the cost of extracting labels using RadBERT compared to five LLMs, or having a human in the loop instead of relying on disagreement between five LLMs.
- Although the authors claim to provide a “solution for extracting reliable reference annotations,” this is not clearly described.
- The authors should consider including the prompts as an appendix in future work. As well as documentation both in the appendix and in the Zenodo repository, especially regarding the annotation protocol, see “Datasheets for Datasets” https://arxiv.org/abs/1803.09010).

[1] Oakden-Rayner, Lauren. "Exploring large-scale public medical image datasets." Academic radiology 27.1 (2020): 106-112.

[2] Oakden-Rayner, Lauren. "Half a million x-rays! First impressions of the Stanford and MIT chest x-ray datasets" https://laurenoakdenrayner.com/2019/02/25/half-a-million-x-rays-first-impressions-of-the-stanford-and-mit-chest-x-ray-datasets/

**Justification Of Rating:**

I recommend accepting this paper. More data work is needed and more discussion about quality of annotations.

---

### Decision · Program_Chairs · 2026-05-08

Accept (Poster)